# Down-Regulation of Rice Glutelin by *CRISPR-Cas9* Gene Editing Decreases Carbohydrate Content and Grain Weight and Modulates Synthesis of Seed Storage Proteins during Seed Maturation

**DOI:** 10.3390/ijms242316941

**Published:** 2023-11-29

**Authors:** Deepanwita Chandra, Kyoungwon Cho, Hue Anh Pham, Jong-Yeol Lee, Oksoo Han

**Affiliations:** 1Kumho Life Science Laboratory, Department of Molecular Biotechnology, College of Agriculture and Life Sciences, Chonnam National University, Gwangju 61166, Republic of Korea; dpllchandra@gmail.com (D.C.); kw.cho253@gmail.com (K.C.); phamhueanh199@gmail.com (H.A.P.); 2Department of Agricultural Biotechnology, National Institute of Agricultural Science, RDA, Jeonju 54874, Republic of Korea

**Keywords:** glutelin, prolamine, seed storage protein, seed quality, starch, protein body, CRISPR-Cas9

## Abstract

The glutelins are a family of abundant plant proteins comprised of four glutelin subfamilies (GluA, GluB, GluC, and GluD) encoded by 15 genes. In this study, expression of subsets of rice glutelins were suppressed using CRISPR-Cas9 gene-editing technology to generate three transgenic rice variant lines, *GluA1*, *GluB2*, and *GluC1*. Suppression of the targeted glutelin genes was confirmed by SDS-PAGE, Western blot, and q-RT-PCR. Transgenic rice variants GluA1, GluB2, and GluC1 showed reduced amylose and starch content, increased prolamine content, reduced grain weight, and irregularly shaped protein aggregates/protein bodies in mature seeds. Targeted transcriptional profiling of immature seeds was performed with a focus on genes associated with grain quality, starch content, and grain weight, and the results were analyzed using the Pearson correlation test (requiring correlation coefficient absolute value ≥ 0.7 for significance). Significantly up- or down-regulated genes were associated with gene ontology (GO) and KEGG pathway functional annotations related to RNA processing (spliceosomal RNAs, group II catalytic introns, small nucleolar RNAs, microRNAs), as well as protein translation (transfer RNA, ribosomal RNA and other ribosome and translation factors). These results suggest that rice glutelin genes may interact during seed development with genes that regulate synthesis of starch and seed storage proteins and modulate their expression via post-transcriptional and translational mechanisms.

## 1. Introduction

Rice (*Oryza sativa*) is a staple food for more than half the world’s population that provides up to 50% of the dietary caloric supply for many of the world’s poorest populations [1]. As a result, rice is one of the most important food crops in the world, especially in Asia. Most widely consumed cereals like wheat, maize, and rice contain seed storage proteins (SSPs) as one of their major components, consisting of approximately 10–12% total seed weight. SSPs fall into the following four categories based on solvent solubility: water-soluble albumins, saline-soluble globulins, alcohol-soluble prolamins, and acid- or alkaline-soluble glutelins [2,3]. Glutelins comprise 60–80% of SSPs in rice, while prolamin is the major SSP in barley, maize and wheat. The rice genome contains 15 glutelin-encoding genes, of which three are pseudogenes, and 12 encode members of the four GluA, GluB, GluC, and GluD subgroups stratified by percent of amino acid sequences similarity [2]. The rice genome also contains 34 prolamin-encoding genes classified into three subgroups stratified by their average molecular weights of 10, 13, and 16 kDa [4]. The prolamins account for 20–30% of rice SSPs [5].

Glutelins are initially produced in the endoplasmic reticulum (ER) as proglutelin precursor proteins, then transported from the ER via the Golgi and stored in Protein Body II (PB-II) vacuoles. Mature glutelins consist of one 37-kDa acidic subunit and one 20-kDa basic subunit connected by disulfide bonds [6,7]. In contrast, mature prolamins remain attached to and surrounded by ER membrane in the ER lumen. Prolamins in the ER aggregate into the 1–2 μm diameter Spherical Protein Bodies I (PB-I) [6,7]; PB-1 vacuoles have an electron-dense central core surrounded by alternating electron-lucent and electron-dense concentric rings [4,6,8,9]. PB-IIs are irregularly shaped, 2 to 4 μm diameter electron-dense structures in which glutelins are stored as a crystalloid lattice in the central core, surrounded by an outer matrix envelope comprised primarily of α-globulin [10].

Starch and SSPs are stored for later use in the rice endosperm during seed development. In maize, development of the endosperm is well understood, but it is relatively poorly studied and not yet well understood in rice. The maize genes that regulate synthesis and storage of starch and SSPs are reported not only to be coordinately expressed but also to function in a concerted manner [11]. Rice starch, which comprises as much as 80% of seed dry weight, serves as a crucial source of carbohydrate and energy during germination and early seedling growth. Rice starch is a significant determinant of both the quality and overall yield of harvested rice. During storage, rice starch is produced and resides in specialized endospermal plastid cells known as amyloplasts [12] and the synthesis of rice starch is directed by an intricate network of regulatory factors [13,14]. At the molecular level, starch is produced through conversion of sucrose into ADP-glucose, generating an endospermal polymer composed primarily of amylose (α-1,4-polyglucans) and amylopectin (α-1,6-branched polyglucans). This process involves four essential enzymes: ADP-glucose pyrophosphorylase (AGPase), soluble and/or granule-bound starch synthase I (SS and GBSSI), branching enzyme (BE), and the debranching enzyme (DBE) [12,15].

Because of their high protein content, long shelf-life, and widespread use as a food staple, rice and other staple grains have potential as a plant-based bioagricultural system for cost-effective oral delivery of therapeutic proteins [16]. Furthermore, in response to the awareness of and demand for healthier food with enhanced protein and nutrient composition, novel approaches to developing engineered food staples seems like an increasingly relevant idea. To that end, we describe here efforts to develop technology for more efficient transgene expression in rice than current technology allows. Our goal is to lay the foundation for successful development of transgenic rice for production and delivery of recombinant and/or therapeutic proteins, and/or to enhance the nutritional value of rice. Importantly, a previous report showed that the yield of recombinant proteins from transgenic rice increased relative to wild-type control rice when a rice variant engineered for low expression of *GluA* and *GluB* was used as the host [17].

Several published studies report characterization of transgenic rice in which SSP expression is knocked down using RNA interference (RNAi) technology. Some of these studies were basic research efforts to understand expression, processing, storage and regulation of other aspects of SSP biology, while others were efforts towards engineering novel rice varieties with reduced protein content (i.e., low-protein rice varieties) [17,18,19,20]. One recent investigation reported use of CRISPR-Cas9 technology to simultaneously inactivate multiple *GluA* and *GluB* genes, thereby generating rice with greatly reduced glutelin content. That study did not specifically examine SSP or starch production, processing or storage in the novel varieties they generated [21]. Here, we report initial characterization of CRISPR-Cas9 engineered rice varieties in which all *GluA*, all *GluB* or all *GluC* genes were selectively inactivated. Seed characteristics including starch and protein composition and configuration of protein storage compartments during endosperm maturation were investigated in these strains, and transcriptomic data were collected, analyzed and compared in immature seeds from the three *glutelin* gene-edited rice varieties. The goal of the latter analysis was to identify the regulatory pathways and relevant determinants of grain composition and weight, because these parameters deviated significantly from wild-type in all three *glutelin* gene-edited varieties. 

## 2. Results

### 2.1. Design of sgRNA for CRISPR-Cas9-Mediated Inactivation of GluA, GluB or GluC Subsets of Rice Glutelin Genes

The 15 rice glutelin-coding genes cluster into four sub-groups (GluA, GluB, GluC, and GluD), each defined by higher relative amino acid sequence similarity within the subgroup than to genes in any of the three other subgroups. Of the 15 rice *glutelin* genes, three are pseudogenes. When the genomic DNA sequences of the 12 *glutelin* genes were aligned, a very high degree of DNA sequence similarity was observed; for example, *GluA1* and *GluA2* demonstrated 93.69% similarity and *GluB1a*/*GluB1b* and *GluB2* demonstrated 89.41% DNA sequence similarity (Figure 1A). The DNA sequences of target genes and the relative abundance of specific *glutelin* gene transcripts (according to the RNA Expression Profile Database: RiceXPro, https://ricexpro.dna.affrc.go.jp/ (accessed on 30 March 2018 )) were used to guide the design of three unique single guide RNAs (i.e., sgRNA-GluA, sg-RNA-GluB, and sgRNA-GluC), each of which could potentially lead to the inactivation of most or all genes in the GluA, GluB and GluC gene subfamilies, respectively. The strategy behind this experiment is presented schematically in Figure 1B. sgRNA-GluA targets a conserved DNA sequence in the fourth exons of *GluA1* and *GluA2*, sgRNA-GluB targets a conserved DNA sequence in the fourth exons of *GluB1a*, *GluB1b* and *GluB2*, and sgRNA-GluC targets a conserved DNA sequence in the third exon of *GluC1*. After the efficacy and specificity of each sgRNA was confirmed by in vitro DNA cleavage assay (Figure 1C), the sgRNAs were cloned into a hybrid expression vector that provided essential functions of both the CRISPR/Cas9 and the pCAMBIA systems, encoded by specific DNA fragments of pRGE31 and pCAMBIA 1201. This generated three binary-vector-derived plasmids, referred as pCAMBIA-Cas9A/B/C, respectively (Figure 1D and Appendix A), which were then used for *Agrobacterium*-mediated transformation of Japonica-type Korean rice cv. *Ilmi*, as described below.

### 2.2. CRISPR/Cas9-Mediated Targeted Editing of All GluA, GluB or GluC Subgroup Rice Glutelin Genes

*Agrobacterium*-mediated transformation of the parental rice variety, Japonica-type Korean rice cv. *Ilmi* using pCAMBIA-Cas9A/B/C yielded 9, 25, and 6 T_0_ transgenic plants, respectively, corresponding to mutant rice varieties *potentially* knocked-down for *GluA1/2*, *GluB1a/1b/2*, and *GluC1*, respectively. However, targeted deep sequencing of genomic DNA from the T_0_ transgenic plants only confirmed the correctly edited *GluA1*, *GluB2*, and *GluC1*-encoding DNA fragments in 2, 10, and 1 T_0_ transgenic plants, respectively (Appendix A & Appendix A). The correctly edited genomic DNA fragments were predicted to encode frameshift mutant variants of the corresponding target genes in 2, 5, and 1 transgenic plants, respectively (Figure 2). More specifically, as shown in detail in Figure 2, *GluA1* was mutated in plants 1a and 2a at frequencies of 99.62 and 24.04%, respectively; *GluB2* was mutated in plants 5b, 7b, 8b, 9b and 10b at frequencies of 50.75, 99.98, 17.84, 50.29 and 48.29%, respectively; and *GluC1* in plant *1c* was mutated at a frequency of 54.42%. Two different mutations were observed in plants 1a, 5b, and 7b, conferring allelic heterogeneity (Figure 2).

### 2.3. Transcriptome Analysis of T_3_ Transgenic Subgroup-Specific Glutelin-Gene Edited Rice Plants

To select homozygous transgenic lines, ten plants from each T_0_ transgenic line were cultivated, and targeted deep-sequencing analysis was conducted until T_2_ generation (Appendix A). In the T_2_ generation, representative homozygous lines (*1a-1-2*, *5b-5-3*, and *1c-2-3*) were carefully chosen for their distinct exhibition of edited gene characteristics, serving as representatives for the knockout of *GluA1*, *GluB2*, and *GluC1* genes. In subsequent studies at T_3_ transgenic generations and beyond, each line was used to evaluate the consequences of *GluA1*, *GluB2*, and *GluC1* gene knockout, respectively. Initial experiments confirmed efficient knockdown of *GluA1*, *GluB2*, and *GluC1* transcripts, respectively, in immature seeds obtained 3 weeks after flowering of the corresponding mutant T_2_ plants (Figure 3A). Subsequently, total SSP extracts from mature seeds of T_2_ plants were analyzed by SDS-PAGE. The results revealed that *1a-1-2* transgenic plants knocked down for *GluA1* and *GluA3* [2,20] expressed low levels of the 35-kDa acidic subunit (black arrow head) in one of the two subclones (Figure 3B). Western blot analysis (Figure 3C) revealed low abundance of GluA1 and GluC1 in the *1a-1-2* and *1c-2-3* lines, respectively, relative to control samples. Western blot data suggested that knockdown of *GluB2* was weak (i.e., poor efficiency of knockdown), possibly reflecting poor antibody specificity and the high amino acid sequence similarity between *GluB1a*, *GluB1b* and *GluB2*. In addition, Western blot data suggested apparent increased abundance of protein products of *GluA3*, *GluB1a*/b, *GluB2*, *GluB4* and *Pro13a* in *1a-1-2* plants, *GluA3*, *GluB4*, *GluC1* and *Pro13a* in *5b-5-3* plants, and *GluA3* in *1c-2-3* plants. This suggests the possibility that decreased abundance of one or more SSPs could trigger compensatory upregulated expression (at the level of translation) of other SSPs.

### 2.4. Morphology of Seeds from T_2_ Glutelin Gene-Edited Transgenic Rice Plants

The seeds of *glutelin* gene-edited T_2_ plants (representing T_3_ progeny) were collected and their morphology and other properties studied. Notably, the shape and size of seeds from T_2_ transgenic plants differed considerably from control wild-type Ilmi seeds. For example: *1a-1-2* seeds were significantly longer and thinner than control seeds (Figure 4); *1c-2-3* seeds were shorter than control seeds; and *5b-5-3* seeds were less thick than wild-type control seeds. Corresponding to their altered morphology, *1a-1-2*, *5b-5-3*, and *1c-2-3* seeds weighed on average as much as 35.81, 17.01, and 14.63% less than wild-type control seeds, respectively (Figure 4B). Under identical physiological conditions, there were no notable morphological distinctions observed between wild-type plants (cv. Ilmi) and *glutelin* gene-edited plants (*1a-1-2*, *5b-5-3*, and *1c-2-3*).

In addition, the endodermal core of these *glutelin* gene-edited seeds had an atypical opaque chalky white color and quality. Consistent with the latter observation, when visualized on a light-emitting background, wild-type Ilmi seeds are translucent while the mutant transgenic seeds from *1a-1-2*, *5b-5-3*, and *1c-2-3* plants were opaque (Figure 5A). Furthermore, scanning electron microscope (SEM) cross-sections of *1a-1-2*, *5b-5-3*, and *1c-2-3* transgenic seed endosperm revealed that starch granules were smaller and more irregular in shape than control seeds, resembling polyhedrons or spheres. The starch granules in the transgenic seeds were also separated by large apparently air-filled spaces, indicating that they were very loosely packed (at least as observed in *1a-1-2*, *5b-5-3*, and *1c-2-3* seeds).Five seeds from both wild-type and each transgenic variety underwent scanning electron microscope evaluation. All seeds of every *glutelin* gene-edited line showed consistent structure of starch granules, as described earlier. Consistent with this, total starch content of *1a-1-2*, *5b-5-3*, and *1c-2-3* seeds was on average 14.36, 7.35, and 6.39% lower than the corresponding values for control Ilmi seeds (Figure 5B) and total amylose content was on average 15.38, 20.51, and 7.69% lower, respectively, than in the wild-type control (Figure 5C). These results suggest that synthesis of starch and formation and packing of starch granules were compromised in T_3_
*glutelin* gene-edited rice seeds, resulting in the opaque chalky white appearance of the transgenic seed endosperm as well as the reduced seed average weight (Figure 4B and Figure 5B).

### 2.5. SSPs and Protein Bodies in T_3_ Glutelin Gene-Edited Transgenic Rice Seeds

Total SSP content of the T_3_ seeds of *glutelin* gene-edited transgenic rice was significantly lower than total SSP content of control Ilmi seeds, with decreasing relative impact on *5b-5-3*, *1a-1-2*, and then least-impacted seeds of *1c-2-3* plants (Figure 6A). To analyze the SSP content of transgenic rice seeds in greater detail, SSPs were selectively extracted using different solvents to yield three SSP fractions: the saline-soluble albumin-globulin fraction, the alcohol-soluble prolamin fraction, and the lactic acid-soluble glutelin fraction (Figure 6B). Subsequent analysis revealed the following: for *1a-1-2* seeds, glutelin:total SSP ratio is lower but glutelin:albumin-globulin and glutelin:prolamin ratios are higher than in control seeds; for *5b-5-3* and *1c-2-3* seeds, glutelin:total SSPs were not significantly different than in control seeds, while the glutelin:prolamin ratio increased slightly. Furthermore, qRT-PCR (Figure 6C) showed atypical patterns of *glutelin* and *prolamin* gene expression (at the level of transcription) during seed development. More specifically, transcriptomic analysis revealed specific differences in transcript abundance in immature seeds of *glutelin* gene-edited rice relative to wild-type Ilmi control seeds. In general, the above analysis suggests that CRISPR-Cas9 knockdown of one of the four subgroups of glutelins has a direct impact on the rice transcriptome, changes relative glutelin and SSP content of rice seeds. Furthermore, these observations strongly suggest that lower expression of targeted glutelins in transgenic rice plants triggered compensatory changes in expression of prolamin, globulin and non-targeted glutelins.

Our next line of investigation focused on the formation, abundance and structure of PB-I and PB-II protein bodies in seeds of transgenic *glutelin* gene-edited rice. As noted above, prolamins and glutelins are stored during development in structures known as PBI and PBII, respectively. Here, the structure and arrangement of PBI and PBII vacuoles in the maturing endosperm was observed by SEM 12 days after the transgenic plants flowered (12 DAF). In wild-type control Ilmi seeds, PB-II vacuoles were irregular in shape and layerless, while PB-I vacuoles appeared to be highly structured layered spheres, where the layers of the PB-I structure reflect distinct electron-dense spherical zones alternately enriched in Pro13aI/II, 13bI/II, 10, and 16 [3,22]. In the seeds of *glutelin* gene-edited rice plants, PB-IIs were on average smaller than in Ilmi control seeds, probably reflecting overall suppressed expression of *glutelins* and low glutelin protein content in the endosperm of transgenic seeds. In addition, the layer structure of PB-I vacuoles was disrupted in *1a-1-2* and *5b-5-3* seeds, but not in *1c-1-3* seeds. This supports the hypothesis that *glutelin* gene knockdown is associated with compensatory changes in prolamin gene expression, a phenomenon that is more clearly demonstrated in *1a-1-2* and *5b-5-3* than in *1c-1-3* transgenic rice (Figure 6D).

### 2.6. RNA-Seq/Global Transcriptome Analysis in Glutelin Gene-Edited Rice Seeds

Global transcriptome profiling of immature (3 WAF) transgenic and wild-type seeds revealed ≥2-fold differential over- or under-expression of 1162 (969 up, 193 down), 2612 (2226 up, 386 down), or 858 (770 up, 88 down) transcripts in *1a-1-2*, *5b-5-3*, or *1c-2-3* transgenic seeds, respectively, relative to wild-type Ilmi seeds. To identify and understand the functional roles and significance of these differentially-expressed genes (DEGs), the genes were assigned to “BIN codes” using the MapCave tool of the MapMen software (see http://mapman.gabipd.org/web/guest/mapcave (accessed on 29 July 2023)) (Appendix A). Table 1 summarizes the results of this analysis, which identified 3364 non-redundant transcripts that were assigned to 28 functional categories based on annotations in the MapMen database including (in order of decreasing % of all transcripts): unknown (59.72%), RNA processing (7.67%), transfer RNA for protein synthesis (4.31%), ribosomal RNA for protein synthesis (3.36%), ribosome, initiation and elongation for protein synthesis (2.47%), regulation of transcription (2.17%), miscellaneous (2.08%), stress (1.93%), protein degradation (1.75%), transport (1.55%), signaling (1.40%), and other minor categories. To validate these results, a subset of differentially expressed transcripts were quantified by qRT-PCR. These two datasets were subject to linear regression analysis. The results reveal robust positive linear correlation between the transcriptomic and qRT-PCR gene expression analyses of DEGs in transgenic rice seeds (R-squared > 0.75; Appendix A), which cross-validates both datasets and both approaches for studying differential gene expression in immature seeds from transgenic rice.

### 2.7. Correlation between Differential Gene Expression and Reduced Rice Seed Weight

The relationship between differential gene expression and average rice seed weight was also investigated using Ilmi, *1a-1-2*, *5b-5-3*, and *1c-2-3* plants. Of 3364 non-redundant DEGs, 801 transcripts were identified for further investigation based on a correlation coefficient ≥ |0.7| (Figure 7). Gene Ontology (GO) functional enrichment analysis suggested roles for these DEGs in ribosome construction and peptidase inhibitor/regulator activity (Appendix A). To explore this further, this subset of 801 DEGs was analyzed using the STRING algorithm, which predicts protein-protein interactions (PPIs) and a corresponding PPI network for a set of genes/proteins. This approach generated a PPI-network 46 nodes and 616 edges. Each node was annotated using the rice Kyoto Encyclopedia of Genes and Genomes (KEGG) Orthology database (https://www.genome.jp/brite/osa00001 (accessed on 29 July 2023)). The data suggests that the biological roles of these DEGs relate to the ribosome, ribosome biogenesis, protein processing in endoplasmic reticulum (ER), and an mRNA surveillance pathway (Figure 8). More generally, the results suggest involvement of post-transcriptional, translation-related, and post-translational mechanisms leading to reduced weight and starch content in seeds from *glutelin* gene-edited rice. 

### 2.8. Starch Synthesis and ER Stress in Glutelin-Related Genes

The results presented above (Figure 5) show reduced average starch content and atypical starch granule morphology in *glutelin* gene-edited rice seeds relative to wild-type rice seeds. Our analysis identified seven DEGs that appear to regulate starch metabolism and six genes appear to positively regulate seed weight and starch content (Figure 9A). The latter DEGs encode starch synthase 3a (SSIIIa), nuclear transcription factor Y subunit B-1 (NF-YB1), floury endosperm 2 (FLO2), sucrose transporter 1 (SUT1), xylose transporter protein homolog 2 (XTPH2), and glucose 6-phosphate/phosphate translocator 1 (GPT1). The gene encoding alpha-amylase isozyme 3A (AMY1.2) correlated negatively with seed weight and starch content.

To confirm these results and their interpretation, transcripts from the following 11 rice genes were quantified by qRT-PCR: ADP-glucose pyrophosphorylase small and large subunits (AGPS2 and AGPL2), granule-bound starch synthase I (GBSSI), starch synthases SSI, SSIIa, and SSIIIa, branching enzymes BEI, BEIIa, BEIIb, plastidial phosphorylase Pho1 and debranching enzymes isoamylase (ISA1) (Figure 9B). Seven genes involved in the synthesis of amylose and amylopectin, *AGPS-2*, *GBSSI*, *SSI*, *SSIIa*, *SSIIIa*, and *BEI*, were significantly down-regulated in *1a-1-2* and *5b-5-3* transgenic rice, while *AGPL-2* was down-regulated in *1a-1-2* and *ISA-1* was down-regulated in *5b-5-3*. *BEI*, *BEIIa*, *Pho-1*, and *ISA-1* were significantly down-regulated in In *1c-2-3*.

Expression of relevant transporters and transcription factors was also analyzed in *glutelin* gene-edited rice. *FLO-2*, *NF-YB-1*, and *FSE-1* transcription factors/regulators were down-regulated in *1a-1-2*, and these genes as well as *NF-YC-12* were down-regulated in *5b-5-3* but none of these genes were down-regulated in *1c-2-3*. Furthermore, sucrose transporter 1 (SUT1), glucose phosphate translocator 1 (GPT1), and ADP-glucose transporter 1 (BT1) were down-regulated in *1a-1-2,* while only *GPT1* was down-regulated in *5b-5-3* and no transporters were down-regulated in *1c-2-3* (Figure 9B).

Figure 9A showed that the expression pattern of nine genes related to protein processing in ER is negatively correlated with the change of grain weight and starch content in *glutelin* gene-edited lines. Based on rice KEGG pathway for protein processing in ER, six genes, except for *CALR* gene, encode proteins that participate in ER-associated degradation of misfolded proteins, indicating that the increase in misfolded protein and degradation in ER (ER stress) may be associated with the decrease in grain weight and starch content. Indeed, to investigate whether genes involved in ER-stress are activated in immature seeds of *glutelin* gene-edited lines, expression levels of eight genes associated in ER stress was examined, including *BiP-1*, *PDIL-1-1*, *PDIL-2-3*, *CNX*, *VPE-1*, *Sar1a*, *Sar1b*, and *Sar1c* (Figure 9B). In the *1a-1-2* line, which showed the most severe decrease in starch content and grain weight, seven genes, except for *VPE-1*, were significantly upregulated compared to Ilmi (Figure 9B). However, in *5b-5-3*, and *1c-2-3*, which showed moderate changes, the genes with a significant change were none except the two (*PDIL-1-1* and *Sar1a*), respectively.

## 3. Discussion

### 3.1. Atypical Morphology of Protein Bodies in the Endosperm of Seeds from Glutelin Gene-Edited Transgenic Rice

ER-derived PB-I vacuoles begin to form in response to the aggregation of β-/γ-zeins (maize) and 10-kDa prolamins (rice) and then expand by co-aggregation of α-/δ-zeins (maize) and 13-/16-kDa prolamins (rice) [3,22,23]. Saito’s group reported that PB-Is have a spherical layer structure consisting of a core region rich in 10-kDa prolamin, an inner layer rich in prolamin 13b, a middle layer rich in prolamin 13a and 16-kDa prolamin, and an outmost layer rich in prolamin 13b [3], thus indicating that the different electron density by the number of cysteine (Cys) in each prolamin results in the layered structure under electron microscopy. In addition to prolamin-rich PB-I particles, glutelins and globulins accumulate and are transported via Golgi to the endosperm, where they form PB-II vacuoles. The globulins self-associate to form an outer spherical frame subsequently filled and enlarged by glutelins [24].

Many published studies report efforts to down-regulate SSPs in rice using RNA-silencing [24], RNA-interference [17,18,19,20,23], and CRISPR/Cas9 [21] technologies, some targeting one or more glutelin-, prolamin-, and globulin- encoding rice genes. The results showed that reduced expression/abundance of the targeted SSPs correlated with increased expression of other SSPs. For example, more abundant glutelins and less abundant prolamins, or the reverse. This led to atypical glutelin:prolamin:globulin ratios as well as PB-I/-IIs characterized by atypical shape, size and/or abundance. The process of grain filling and endosperm development in rice remains not fully explored and characterized. The intricate biosynthetic pathways of various seed storage proteins (SSPs) make it challenging to predict the balance in rice endosperm. Further investigations are needed to fully understand how the knockdown or knockout of a specific glutelin can induce compensatory alterations in the expression of other glutelins, prolamin, and globulin in rice. Similarly, in this study, we observed the reduction of total SSPs and suppression of targeted SSPs in the seeds of *glutelin* gene-edited lines, together with the enhanced expression of other *glutelin* genes and increased number of smaller and irregular PB-IIs. However, phenotypes of PB-Is varied depending on the *glutelin* gene-edited line. In immature seeds of the *GluB2* gene-edited line, Cys-rich 10-kDa prolamin in the core region of PB-I, and prolamin 13a at the middle layer were increased at transcript or protein levels, while Cys-poor prolamin 13b at the electron-lucent layers was decreased, resulting in forming small and dark PB-Is (Figure 6D). In contrast, in *1a-1-2* seeds, prolamin gene expression was unperturbed, but PB-I vacuoles appeared to be “cracked”, while in *1c-1-3* seeds, PB-I vacuoles resembled the wild-type control particles (Figure 6D). PB-I structures with a ‘cracked’ appearance were previously reported in a study of plants in which the pathway for endoplasmic reticulum-associated degradation (ERAD) was strongly down-regulated [25]. The authors of the study hypothesized that the ‘cracks’ they observed in PB-I structures represent an accumulation of self-aggregating damaged and/or mis-folded prolamins.

ERAD is a protein-degradation pathway that eliminates mis-folded, damaged and unfolded proteins to protect the integrity of the proteome. In plants, ERAD is required to ensure that only functional, undamaged SSPs accumulate in the endosperm, and it acts largely on highly expressed SSPs during intense periods of SSP biosynthesis in immature seeds [17,26,27]. The ERAD is processed through four major steps: recognition, ubiquitination, retro-translocation, and degradation. Knock-down of Hrd3, which participates in the recognition process, induces the accumulation of unfolded RM1 (pro13a.2) proteins in the ER lumen, leading to ER-stress [28]. Knock-down of DER1, a protein- interacting partner of Hrd3, did not significantly impact biosynthesis of SSPs but was associated with appearance of ‘cracked’ PB-I vacuoles resembling those observed in *GluA1* gene-edited rice [25]. Suppression of endoplasmic reticulum-resident J-protein7 (ERdj7), which participates in the translocation of SSPs within the ER lumen and the degradation of misfolded proteins in rice by interacting with Hrd3, results in abnormal accumulation of Cys-poor prolamin 13b and Cys-rich prolamin 13a in ER lumen, forming atypical PBs with mesh-like structures [28,29,30]. Consistent with this, among transcripts that were down-regulated in *1a-1-2* and *5b-5-3* (but not *1c-1-3*) rice variants, the third most highly enriched subgroup was assigned to GO category ‘protein degradation.’ These transcripts encoded ubiquitin ligases, proteases and other proteins involved in protein degradation.

### 3.2. Differential Gene Expression in Transgenic Rice Correlated with Low Seed Weight and Reduced Seed Starch Content

During the maturation of seeds, synthesized SSPs are transported to morphologically distinct PBs from the ER lumen through a mechanism facilitated by ER chaperones. An excessive build-up of unfolded proteins in ER lumen triggers ER stress and initiates the unfolded protein response, leading to an up-regulation of ER chaperone genes such as *BiP-1*, *PDIL1-1*, *PDIL2-3*, *CNX*, *VEP-1*, and *Sar1a/1b/1c*, and ensuring the preservation of ER equilibrium [31]. BiP1 promotes prolamin folding and its assembly into PB-I in rice endosperm [32]. PDILs (protein disulfide isomerase-like proteins) form intra- or intermolecular disulfide bonds in glutelins and prolamins to promote maturation and proper folding of SSPs during seed maturation. PDIL1-1 is involved in the oxidative folding of proglutelins [17,33,34], whereas PDIL2-3 promotes the assembly of Cys-rich 10-kDa prolamin in PB-I formation [34]. Sar1, small GTPase Secretion-associated and Ras-superfamily-related1, is a component of coat protein complex II (COPII) involved in ER-to-Golgi transport and plays a crucial role in forming ER-derived spherical PBs containing proglutelin and α-globulin [35,36]. VPE1 (vacuolar processing enzyme 1) has a cysteine protease activity, which cleavages proglutelin into acidic and basic subunits when proglutelin is transported from Golgi to PB-IIs [37,38]. This supports the idea that increased expression of *BiP-1*, *PDIL1-1*, *PDIL2-3*, and *Sar1b/1c* causes serious ER-stress; as a result, *1a-1-2* transgenic seeds display signs of much higher ER stress than *5b-5-3*, and *1c-1-3* seeds, because all four genes are upregulated in *1a-1-2*, only one of these genes is up-regulated in *5b-5-3* seeds and none are upregulated in *1c-1-3*.

Among *1a-1-2*, *5b-5-3*, and *1c-1-3* lines, grain weight and starch content in seeds varied according to lines. These phenotypic changes correlated with increased or decreased abundance of transcripts involved in ribosome and peptidase inhibitor activity (Appendix A). More specifically, KEGG pathway analysis and functional protein network analysis using STRING showed that proteins categorized in protein processing in ER, ribosomal small and large subunits, ribosome biogenesis, and mRNA surveillance pathway establish a protein–protein interaction network (Figure 8). The mRNA surveillance pathway controls mRNA quality during post-transcriptional or translational processing by nonsense-mediated decay (NMD) for degradation of mRNA with premature stop codon, nonstop decay (NSD) for degradation of mRNA without stop codon, and no-go decay (NGD) for degradation of mRNA with translational stalling [39,40]. In this study, *GluA1*, *GluB2*, and *GluC1* genes were edited using the CRISPR/Cas9 system, resulting in InDel mutation at one exon of each gene, which appears to have reading frame shifts and premature stop codons (Appendix A). The negative correlation between genes involved in mRNA quality control and grain quality may be due to a basal level of *GluA1* gene expression during seed maturity being higher than those of *GluB2* and *GluC1* genes. In other words, eliminating mRNAs with a premature stop codon in *GluA1* gene-edited (*1a-1-2*) line would have required more mRNA surveillance than in other lines.

Additionally, in the network, there are seven proteins implicated in protein processing in ER, including five heat shock proteins (two HSP20s, two HSP90As, and HSP90B), protein disulfide-isomerase A1 (PDIA1), and calreticulin (CALR). HSP20 and HSP90 interacting with many co-chaperones join in ERAD, a complex process through which misfolded proteins are selected and ultimately degraded by the ubiquitin-proteasome system [41,42]. CALR is a highly conserved chaperone protein in the endoplasmic reticulum, and is involved in protein folding quality control and calcium homeostasis [43,44,45,46]. PDIA1 interacting with ERO1 recognizes unfolded and misfolded proteins to mediate ERAD [47]. HSP90B interacting with OS-9 delivers aberrant proteins to ubiquitin ligase complex for ERAD [48]. The compositional analysis of SSPs showed that the proportion of glutelin in *1a-1-2* line was most severely disrupted compared to those in *5b-5-3* and *1c-1-3* lines, which are similar to Ilmi. The analysis also suggested that ERAD-related gene expression was strongly upregulated in the *1a-1-2* variant, in which *GluA1* gene was edited by CRISPR-Cas9, reflecting decreased glutelin content, increased ER stress and strong upregulation of ERAD.

In the *glutelin* gene-edited lines, the expression of numerous genes associated with storage starch production was significantly diminished (Figure 9B). Moreover, the structure and appearance of starch granules within the mutant endosperms differed markedly from those in the wild-type. Phenotypically, the seeds of the mutant lines exhibited a chalky and shrunken appearance, which was notably distinct from the wild-type seeds (cv. *Ilmi*), as depicted in Figure 4 and Figure 5. Starch biosynthesis is initiated from the conversion of glucose 1-phosphate and ATP to ADP-glucose by ADP-glucose pyrophosphorylase, a complex of small subunit (AGPS) and large subunit (AGPL) [49,50,51]. The mutant rice variants demonstrated significantly reduced cytosolic ADP-glucose pyrophosphorylase activity, leading to low starch content and enlarged starch granules [52]. Amylose is primarily produced through the action of granule-bound starch synthase I (GBSSI) [53], amylopectin is mainly elongated by soluble starch synthases (SSSs), starch branches are formed by starch branching enzymes (SBE) by transferring an oligosaccharide fragment, and the branches are broken by starch phosphorylase (Pho1) and isoamylase 1(ISA1) [54,55,56]. The rice grains in SSIIIa-deficient mutants exhibited starch granules with round shape, smaller size, and less crystalline. The absence of the SSIIIa isozyme was accompanied by increased expression of both SSI and GBSSI genes and resulted in a dual impact on the structure of amylopectin, amylose content, and the physicochemical characteristics of starch granules [57]. Thus, it seems likely that the reduced starch content of glutelin-gene edited mutants *1a-1-2*, *5b-3-3* and *1c-1-3* reflects decreased expression of starch synthesis-related genes (Figure 5C and Figure 9B). 

Mutation in FLOURY ENDOSPERM2 (FLO2) in rice led to diminished grain size and starch quality. Conversely, the over-expression of FLO2 led to a substantial increase in grain size, indicating that FLO2 has a crucial regulatory function in determining rice grain size and starch quality by influencing the accumulation of storage substances in the endosperm [58]. Additionally, the starch synthesis is regulated by the interaction between NF-YB1and OsMADS14, which up-regulates the expression of OsAGPL2 and Wx (GBSSI) genes. The impaired function of OsMADS14 changed the expression of starch biosynthesis genes, which led to reduced starch production and a decline in the quality of the rice endosperm, resulting in producing both shrunken and chalky grains [59]. NF-YB1 has the capacity to directly bind itself to the “CCAAT boxes” present in promoter of sucrose transporter 1 (OsSUT1), thereby activating their expression. This action governs the transport of sucrose to the developing endosperm, facilitating the process of starch production [60]. NF-YB1 is also recognized for its ability to engage with NF-YC12, resulting in the formation of a transcription complex. This complex directly attaches to the “G box” or “GCC box” within the promoters of genes responsible for starch synthesis as well as the transport of sugar and amino acids [61,62,63]. FLOURY SHRUNKEN ENDOSPERM1 (FSE1) is a gene that plays a role in regulating starch biosynthesis indirectly, and it also has an impact on starch quality and grain filling [64]. In the development of rice grains, OsSUT1 has a crucial function in transporting sucrose from the phloem to the filial tissue [65]. A Glucose 6-phosphate/phosphate translocator (GPT) in the amyloplast envelope facilitates the import of glucose 6-phosphate, which can then be utilized for starch biosynthesis in non-green plastids of rice [66]. An ADP-glucose transporter (OsBT1) is situated within the membrane of the amyloplast envelope and serves a vital role in controlling both starch biosynthesis and the creation of complex starch granules throughout the development of rice seeds [67]. In *glutelin* gene-edited lines, the transcription of positive regulators involved in starch metabolism, such as FLO2, NF-YB1, NF-YC12, and FSE1, showed that their down-regulation differs depending on each line (Figure 9B), such that all genes are significantly down-regulated in *5b-5-3* line, three except for NF-YC12 gene in *1a-1-2* line, but none in *1c-1-3* line. The different combination of down-regulated TFs among *glutelin* gene-edited lines determines transcription levels of genes related to starch synthesis and transporters, resulting in the decrease in their starch contents and changes in amylose contents in each line.

## 4. Materials and Methods

### 4.1. Designing gRNA for Different Glutelin Genes

CRISPR-P 2.0, an online tool for guide RNA designing was used. (http://crispr.hzau.edu.cn/CRISPR2 (accessed on 30 March 2018)). This tool provides options to select the organism of choice and their genome database. After selection, the accession number of the gene of interest was entered. gRNAs are designed such that it targets site of high homology between the *glutelin* genes, thus one gRNA is targeting more than one gene. In addition, the designed sgRNAs start with adenine (A) as U3 promoters in plants have a discrete transcription start site with adenine (A) [68]. Three sgRNAs were designed for targeting different *glutelin* genes, named as- sgRNA-GluA, sgRNA-GluB & sgRNA-GluC.

### 4.2. Testing the Efficacy of sgRNAs In Vitro Using Takara Guide-it™ sgRNA In Vitro Transcription and Screening System

A DNA template was generated that contains the designed sgRNA-encoding sequence under the control of a T7 promoter by performing a PCR reaction with the kit included a Guide-it Scaffold Template and a primer designed such as it contains sgRNA sequence and T7 promoter (Appendix A). This template was in vitro transcribed with the included Guide-it T7 Polymerase mix to synthesis a sgRNA. After digestion with Recombinant DNase I (RNase-Free), sgRNA was purified using the Guide-it IVT RNA Clean-Up Kit. A portion of target gene was amplified using cDNA, in such a way that it contained the target sequence in an asymmetric position, which would produce two cleaved fragments of unequal sizes. A mix of target DNA, sgRNA & Cas9 was prepared and incubated at 37 °C for 1 h for cleavage (Cat.#632638, 632639, 632635).

### 4.3. Cloning of Two Different Cas9-sgRNA Vectors

#### 4.3.1. Using NEB DNA Assembly System sgRNAs Are Introduced into pRGE31-Cas9

A 70-base, single-stranded DNA oligonucleotide was designed, containing a 20 nt sgRNA sequence, flanked by a partial U3 promoter sequence and scaffold RNA sequence of pRGE31 vector. 20 µL of reaction mix was prepared with 5 µL of ssDNA oligo (0.2 µM), 30 ng of restriction enzyme-linearized vector and 10 µL of NEBuilder HiFi DNA Assembly Master Mix (Cat.# E2621) and rest ddH2O. The assembled reaction mixture was incubated for 1 h at 50 °C for ligation. 5µL of assembled product was used for transformation into (DH5-α strain) *E. coli* cells. Few colonies were picked to grow, and the plasmid DNA was purified for sequencing.

#### 4.3.2. Cas9 Binary Vector: Cloning *Cas9* Gene and gRNA Scaffold from pRGE31 to pCAMBIA

Double digestion of pCAMBIA 1201 with *Hind III* and *BstE II* was performed, while the large fragment of pCAMBIA 1201 was purified by gel extraction. PCR amplification of Cas9 gene, sgRNA and gRNA scaffold was performed using pRGE31-sgRNA vector as template, with reverse primer having *BstE II* site and forward primer. Before ligation, the PCR product was subjected to double digestion with the same restriction digestion enzymes (*Hind*III and *Bst*EII) and then purified using a PCR purification kit (QIAquick PCR Purification Kit (Cat#28106), Hilden, Germany). The pCAMBIA 1201 vector and the PCR amplified Cas9 gene and gRNA scaffold were ligated using T4 DNA ligase at 16 °C overnight. Positive vector construction was confirmed with colony PCR and DNA sequencing.

### 4.4. Agrobacterium-Mediated Transformation of Rice Callus

The pCAMBIA-Cas9-sgRNA vectors were introduced in *Agrobacterium tumifaciens* (strain AGL1) separately, and then embryogenic calli generated from mature seeds of Korean rice cultivar *Ilmi* (japonica-type) were transformed. The transformation of embryogenic rice calli was performed according to the method of Kim et al., (2012) [17].

### 4.5. Sequencing of Glutelin Genes of Potential Transgenic Plants

A novel method called deep-sequencing was employed for the sequencing of potential transgenic plants. In this technique, each DNA strand in the reaction mix is sequenced independently. To prepare the deep-sequencing template three sets of PCR amplifications were done as instructed by the institute (KAIST Bio Core Center) (Appendix A). After purification, the deep-sequencing of PCR product was performed and then analyzed using a RGEN tool (CRISPR RGENT tools / Cas-Analyzer) (http://www.rgenome.net/ (accessed on 9 January 2023)). Frequency was calculated based on the count of normal sequenced DNA strands and those with the presence of mutations in the specified location (upstream of the PAM sequence).

### 4.6. Phenotypic and Physiological Characterization of Transgenic Seeds

Grain weight of 100 seeds of each transgenic plant and wild-type plant were measured and three independent readings were taken for each line. To measure the length and width of the transgenic seeds, a digital caliper gauge was used. For the measurement of the seed size, 30 seeds were randomly selected and a mean and standard error was calculated. 50 seeds were surface sterilized with 5% sodium-hypoclorite (Daejung, Republic of Korea; Cat.#7681-52-9) and soaked in 10 mL of distilled water in petri-dish for continuous period of 6 days. Germination rate was measured after every 12 h. Germination assay was repeated 3 times then a mean was taken. Field grown wild-type (cv. *Ilmi*) seeds, in same condition as transgenic seeds, were used as the control.

### 4.7. SDS-PAGE and Western Blot Analysis

The total seed storage protein was extracted from 3 mature dry grains from wild-type and *glutelin* gene-edited lines. The rice grains were de-husked and carefully finely grounded into powder using a mortar and pestle, then 1 mL of sodium dodecyl (SDS) urea buffer [250mM Tris-HCl, pH 6.8, 4% SDS, 8 M urea and 20% glycerol] was added and incubated for 3h at RT. A supernatant containing total seeds storage protein was collected and concentration of protein was measured by BCA method (Bicinchoninic Acid Kit for Protein Determination, Cat.#B9643, C2284, Sigma-Aldrich, Saint Louis, MO, USA). 7.5 µg of the extracted protein was separated by 12% SDS-PAGE. After the separation of the protein, the gel was stained with CBB-R250 or the unstained gel was transferred to PVDF membranes for western blot analysis.

### 4.8. RNA Extraction and Real Time qRT-PCR

Immature developing seeds from wild-type and *glutelin* gene-edited lines were harvested after 3 weeks after flowering (WAF). The total RNA was extracted from 3 de-husked seeds using the RNA extraction protocol for seeds containing high level of starch [69]. The cDNA library preparation was done using 1µg of RNA with QuantiTect Reverse transcription Kit (Cat.#205311, Qiagen, Hilden, Germany) according to the manufacturer’s protocol. The q-RT-PCR mix consisted of 6 ng cDNA, 0.5 µM forward primer, 0.5 µM reverse primer, and 10 µL of 2X qPCR Master Mix (QuantiTect SYBR Green PCR Kit, Cat.#204343, Qiagen, Hilden, Germany) in a total volume of 20 µL. The thermal cycling parameters of Qiagen Rotor-Gene Q cycler is as follows: an initial denaturation at 95 °C is conducted for 10 min, followed by 40 PCR cycles at 95 °C for 30 s, 60 °C for 30 s, 72 °C for 30 s, and a melting curve analysis from 72 to 95 °C. The expression levels of each gene were normalized using a housekeeping gene Ubiquitin 10 (*LOC_Os02g06640*). RT-PCR data were analyzed using the 2^−ΔΔCT^ method [70]. In addition, expression levels of each gene in WT were normalized to 1 to facilitate better comparison (Appendix A).

### 4.9. Fractionation of Seed Storage Proteins (SSP)

Fractionation of SSP in rice was performed according to the method of Lee et al., 2015 [19] with some modifications as follows. Ten rice grains were finely ground into powder with a mortar and pestle. The rice flour was defatted with 500 μL of acetone at 4 °C for 30 min. After centrifugation at 13,000× *g* for 10 min, the supernatant was discarded. The pellet was dissolved in a PBS buffer (PBS; 0.866 M K2HPO4, 0.134 M KH2PO4, and 0.4 M NaCl, pH 7.5); the solution was incubated at RT for 4h with constant shaking. The supernatant was collected containing albumin and globulin. The last step was repeated and a second fraction of albumin and globulin was collected. Next, the pellet was dissolved in 60% isopropanol incubated in a shaker for 4 h and centrifuged to collect prolamin; repeated to collect a second fraction of prolamin. The pellet was washed with ddH2O and dissolved in 1% lactic acid solution containing 1mM EDTA-2Na and incubated for 4 h in shaker; centrifuged to obtain glutelin. Similarly, another fraction of glutelin was collected.

### 4.10. Starch and Amylose Content

Rice flour was prepared by finely grinding ten seeds each of wild-type and *glutelin* gene-edited lines. Equal quantities (25 mg) of the rice flour were then measured and utilized to assess starch and amylose content. The determination of starch content and amylose content was conducted using a Amylose/Amylopectin Assay Kit (Megazyme, Cat.#K-AMYL) and following the procedural guidelines provided by the manufacturer.

### 4.11. Transcriptome Data Analysis

Immature grains (3 WAF) were collected from field grown rice plants of wild-type (cv. *Ilmi*) and *glutelin* edited lines. RNA was extracted from each sample and sent to DNALINK biotechnology company (Seoul, Republic of Korea) for quality checks, cDNA library construction and primary data analysis. In the process of data analysis, Kallisto tool was utilized to align the clean reads from each sample with the rice reference genome sequence (Nipponbare rice genome IRGSP-1.0). (https://pachterlab.github.io/kallisto (accessed on 11 July 2023)) [71]. Quantification of each transcript is established using a tool named RSEM v1.3.3 (http://deweylab.biostat.wisc.edu/rsem/ (accessed on 11 July 2023)) [72]. Differentially expressed gene analysis was performed using the DESeq2 method on edgeR software (https://bioconductor.org/packages/release/bioc/html/edgeR.html (accessed on 11 July 2023)) [73,74].

### 4.12. Selection of Transcripts Correlated with Grain Quality and Establishment of PPI Networks

To investigate transcripts linked with grain quality, we performed a Pearson correlation test between the expression levels of differentially expressed transcripts and grain weight in *glutelin* gene-edited rice and wild-type, and selected transcripts with correlation coefficient absolute value greater than 0.7. The selected transcripts were annotated according to database of Gramene Mart of Gramene (https://www.gramene.org/ (accessed on 11 July 2023)) linked with *Oryza sativa* Japonica group genes (IRGSP-1.0), Osa_RAPDB (Rice Annotation Project Database) reference map of *Oryza sativa* Japonica goup (RAPDB-IRGSP1.0) in MapMan (https://mapman.gabipd.org/home (accessed on 11 July 2023)), and KEGG Orthology (KO)-*Oryza sativa* japonica (Japanese rice) (RefSeq) (https://www.genome.jp/brite/osa00001 (accessed on 11 July 2023)). A PPI (protein-protein interaction) network was analyzed by using the STRING web-based tool (https://string-db.org/cgi (accessed on 11 July 2023)) set to a confident interaction score ≥ 0.4 to observe the relationship among the selected transcripts. The PPI network was visualized using Cytoscape software (ver. 3.4.0) with output data of the STRING tool. To prepare the heat-map, candidate genes involved in starch metabolism and chaperons were selected from the 801 correlated genes, their expression pattern among different lines were normalized by z-score and visualized using PermutMatrix software (ver. 1.9.3) set to as Manhattan distance dissimilarity and hierarchical clustering linked to ward’s minimum variance method (Appendix A).

### 4.13. Scanning Electron Microscope (SEM)

To observe the structure of starch granules in the endosperms, rice seeds were cut across the short axis with a sharp razor blade. The samples were mounted on SEM stubs and coated with platinum particles. The mounted specimens were examined by a scanning electron microscope (SEM) in a secondary electron mode at 15kV.

### 4.14. Transmission Electron Microscope (TEM)

Immature rice seeds (12 DAF) were fixed in a mixture of 2% glutaraldehyde and 2% paraformaldehyde in 0.05M cacodylate buffer (pH 7.2) for 4 h at room temperature. After three buffer washes at 30-min intervals, they were fixed in 1% OsO4 for 1 h, followed by three buffer washes. Dehydration was carried out using a graded ethanol series (30–100%) at 30-min intervals, and samples were embedded in LRWhite resin at 60 °C for 24 h. Ultrathin sections (80–100 µm thick) were prepared, mounted on carbon-coated nickel grids, double-stained with 4% uranyl acetate and 0.4% lead citrate, and examined using a transmission electron microscope.

## Figures and Tables

**Figure 1 ijms-24-16941-f001:**
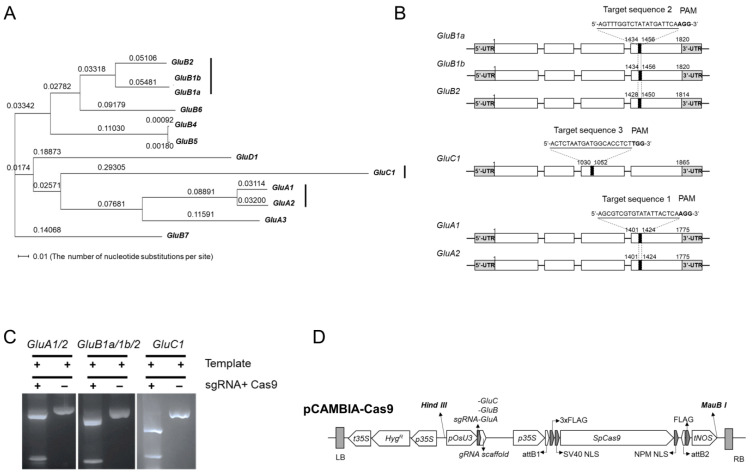
Phylogentic tree of glutelin genes and design of conservatively distinct sgRNAs for edit glutelin genes. (**A**) Phylogenic analysis of rice glutelin genes. Rectangular phylogenic tree is established by using D (ver.3.5.7) after the alignment by using a web-based Clustal Omega tool (https://www.ebi.ac.uk/Tools/msa/clustalo/ (accessed on 9 Ausust 2023)) with genomic DNA sequences (from start codon to stop codon) of rice glutelin genes such as GluA1 (Os01g0762500) GluA2 (Os10g0400200), GluA3 (Os03g0427300), GluB1a (Os02g0249800), GluB1b (Os02g0249900), GluB2 (Os02g0249600), GluB4 (Os02g0268300), GluB5 (Os02g0268100), GluB6 (Os02g0248800), GluB7 (Os02g0242600), GluC1 (Os02g0453600), and GluD1 (Os02g0249000). (**B**) Three conservatively distinct sgRNAs, target sequence 1 for GluA1 and GluA2, target sequence 2 for GluB1a, GluB1b, and GluB2, target sequence 3 for GluC1. Black boxes indicate specific target sites of the designed sgRNA for each gene editing. White boxes and lines present exons and introns of each gene, respectively. UTR means untranslated region. (**C**) In vitro cleavage assay of sgRNAs designed for editing glutelin genes. After amplification of genomic DNA fragments containing target sequences of glutelin genes by PCR, the products are cleaved with synthesized sgRNAs and recombinant Cas9 nucleases according to manufacture instruction (Guide-it Complete sgRNA Screening System, Cat. No. 632636), and then separated by agarose gel (1.2%) electrophoresis. (**D**) Construction of a CRISPR/Cas9 binary vector to edit the rice glutelin genes.

**Figure 2 ijms-24-16941-f002:**
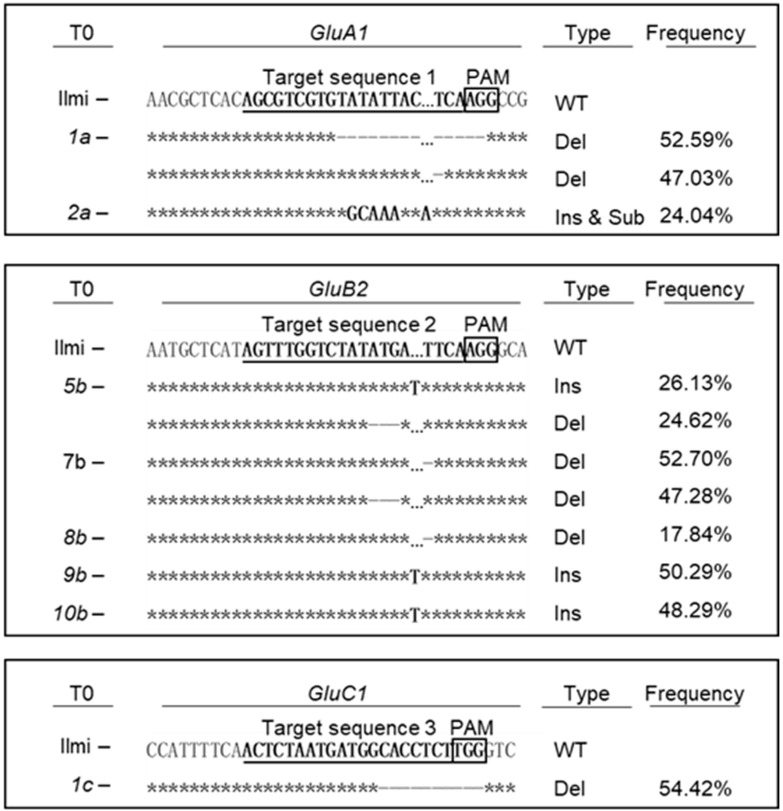
Sequence analysis showing T_0_ plants with frame shift mutations in their respective target gene. CRISPR-Cas9 target sequence is underlined and PAM sequence is boxed. Based on deep sequencing analysis, indel frequency of each line is predicted. ‘*’ represents nucleotides identical to those in the original wild-type sequence.

**Figure 3 ijms-24-16941-f003:**
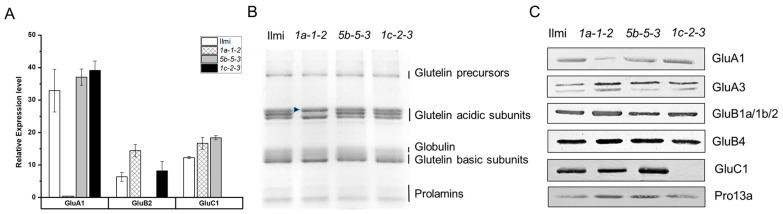
Expression analysis of *glutelin* genes at transcript and protein levels in wild-type (cv. *Ilmi*) and *glutelin* gene-edited lines (*1a-1-2*, *5b-5-3*, and *1c-2-3*). (**A**) Relative expression level of *GluA1*, *GluB2*, and *GluC1* genes in wild-type and *glutelin* gene-edited lines. q-RT-PCR was performed to detect the transcript levels of mutated genes in immature seeds at 3 WAF of wild-type and *glutelin* gene-edited lines (*1a-1-2*, *5b-5-3*, and *1c-2-3*). (**B**) Comparison of SSPs of WT and *glutelin* gene-edited lines on SDS-PAGE (12%). (**C**) Western blot analysis using anti-SSP antibodies with total seed storage protein extracts derived from wild-type and *glutelin* gene-edited lines (*1a-1-2*, *5b-5-3*, and *1c-2-3*).

**Figure 4 ijms-24-16941-f004:**
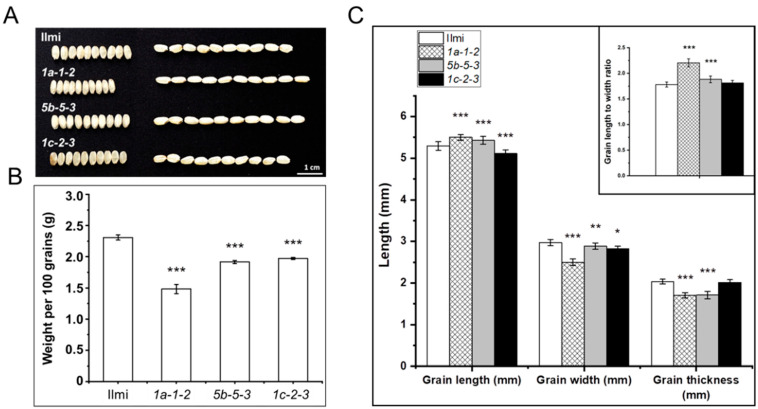
Comparison of grain morphology and grain shape of *glutelin* gene-edited lines (*1a-1-2*, *5b-5-3*, and *1c-2-3*) with wild-type (cv. *Ilmi*). (**A**) The phenotypic comparison of wild-type (cv. *Ilmi*) and *glutelin* gene-edited seeds (*1a-1-2*, *5b-5-3*, and *1c-2-3*). (**B**) Weight of 100 grains of wild-type (cv. *Ilmi*) and *glutelin* gene-edited lines (*1a-1-2*, *5b-5-3*, and *1c-2-3*). (**C**) Statistical analysis of grain length (mm), grain width (mm), grain thickness (mm) and length to width ratio of wild-type (cv. *Ilmi*) and *glutelin* gene-edited lines (*1a-1-2*, *5b-5-3*, and *1c-2-3*). Error bars denote ± SD of three replicates. *p* values were calculated by the Student’s *t* test (* *p* < 0.1, ** *p* < 0.01 and *** *p* < 0.001).

**Figure 5 ijms-24-16941-f005:**
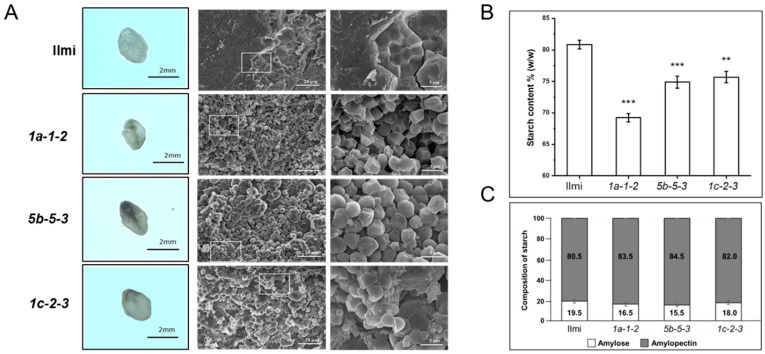
Endosperm appearance, starch granule and total starch content in seeds of wild-type and *glutelin* gene-edited lines (*1a-1-2*, *5b-5-3*, and *1c-2-3*). (**A**) Comparative observation of opaqueness in transverse section of seeds and structure of starch granules. Seeds were cut in thin sections and photographed on a light emitting background. Scanning electron microscope (SEM) was used to observe endosperms of wild-type (cv. *Ilmi*) and *glutelin* gene-edited lines (*1a-1-2*, *5b-5-3*, and *1c-2-3*). (**B**) Total starch content content (%, *w*/*w*) in seeds of wild-type (cv. *Ilmi*) and *glutelin* gene-edited lines (*1a-1-2*, *5b-5-3*, and *1c-2-3*). (**C**) Composition of starch (amylose% and amylopectin%) in seeds of wild-type (cv.Ilmi) and *glutelin* gene-edited lines (*1a-1-2*, *5b-5-3*, and *1c-2-3*). Error bars denote ± SD of three replicates. *p* values were calculated by the Student’s *t* test (** *p* < 0.01 and *** *p* < 0.001).

**Figure 6 ijms-24-16941-f006:**
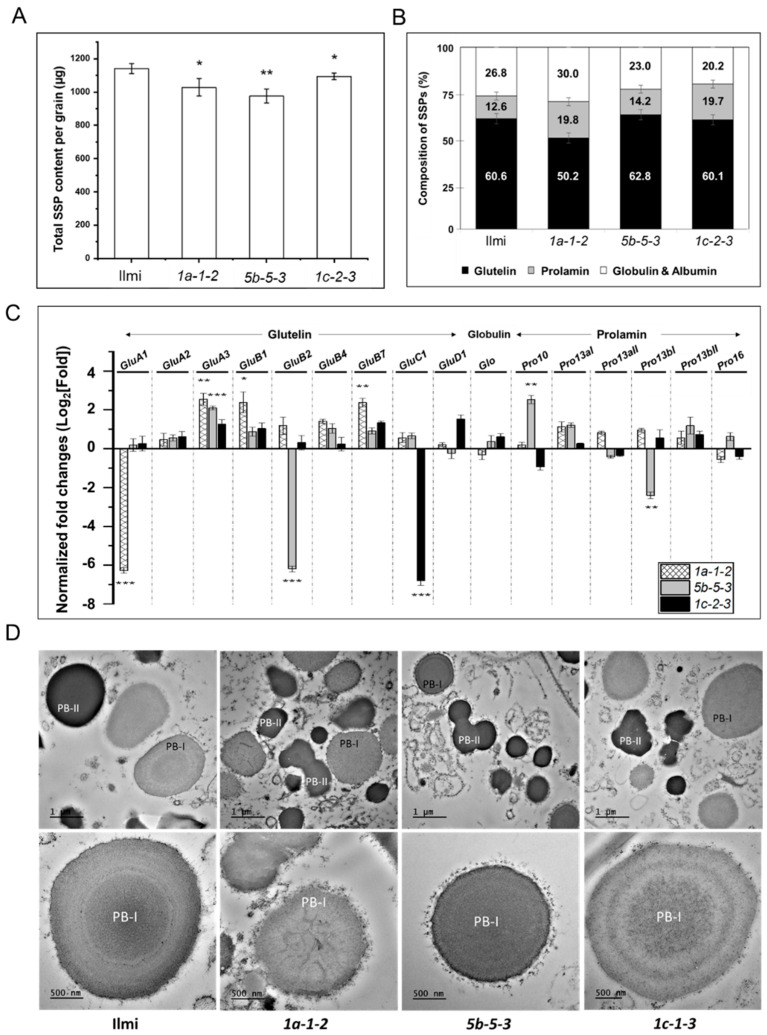
Composition of SSPs and structure of protein-bodies in wild-type (cv. *Ilmi*) and *glutelin* gene-edited lines (*1a-1-2*, *5b-5-3*, and *1c-2-3*) (**A**) Amount of total SSP per grain (µg/grain). (**B**) Comparison of percentage of SSP fractions (albumin, globulin, prolamin and glutelin) in wild-type (cv. *Ilmi*) and *glutelin* gene-edited seeds (*1a-1-2*, *5b-5-3*, and *1c-2-3*). (**C**) Quantitative real-time PCR analysis of SSP genes in *glutelin* gene-edited lines. Transcript levels of different seed storage protein genes were analyzed in the immature seeds (3 WAF) of wild-type (cv. *Ilmi*) and *glutelin* gene-edited lines (*1a-1-2*, *5b-5-3*, and *1c-2-3*). Normalization of expression levels of the target genes was performed using the 2^−ΔΔCT^ method and the relative level of individual gene in each glutelin gene-edited lines versus wild-type is represented as log2 (average fold change) value. (**D**) Phenotypic characterization of protein bodies in the developing endosperm. Transmission electron microscope was used to observe the protein bodies in developing endosperms (12 DAF) of wild-type (cv. *Ilmi*) and *glutelin* gene-edited lines (*1a-1-2*, *5b-5-3*, and *1c-2-3*). PB-I: protein body I; PB-II: protein body II. Error bars denote ± SD of three replicates. *p* values were calculated by the Student’s *t* test (* *p* < 0.1, ** *p* < 0.01 and *** *p* < 0.001).

**Figure 7 ijms-24-16941-f007:**
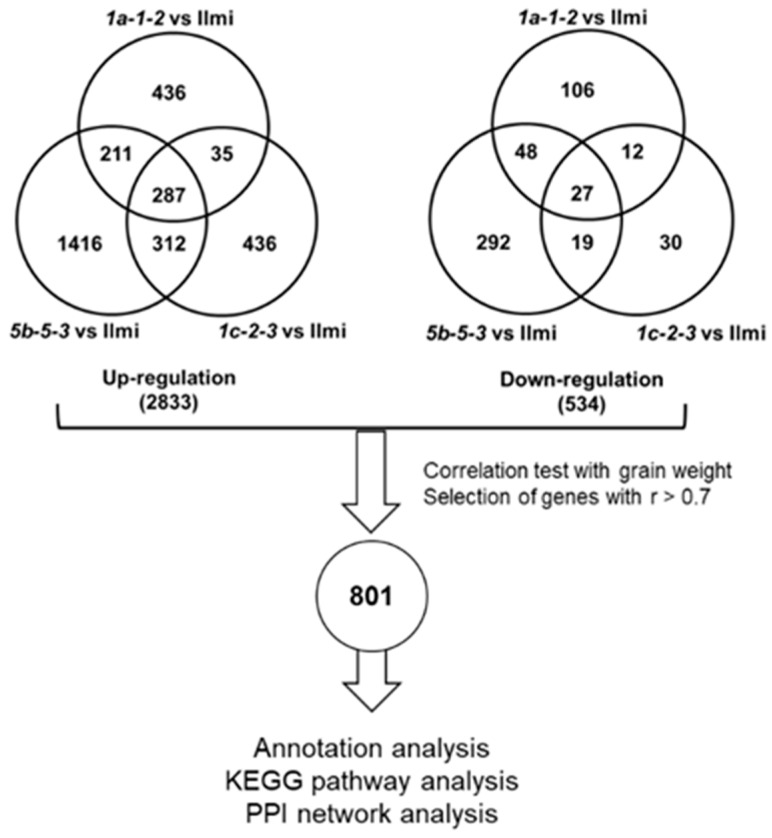
Transcriptome analysis and correlation analysis with DEGs in *glutelin* gene-edited lines. Venn diagrams show a total of 2833 upregulated genes and 534 down-regulated genes in *glutelin* gene-edited seeds when compared with wild-type (cv. *Ilmi*) seeds at 3 WAF. Correlation assay was carried out between changes in grain weight and differentially expressed genes in *glutelin* gene-edited lines. 801 transcripts having at least 0.7 absolute value of correlation coefficient were selected for further analysis.

**Figure 8 ijms-24-16941-f008:**
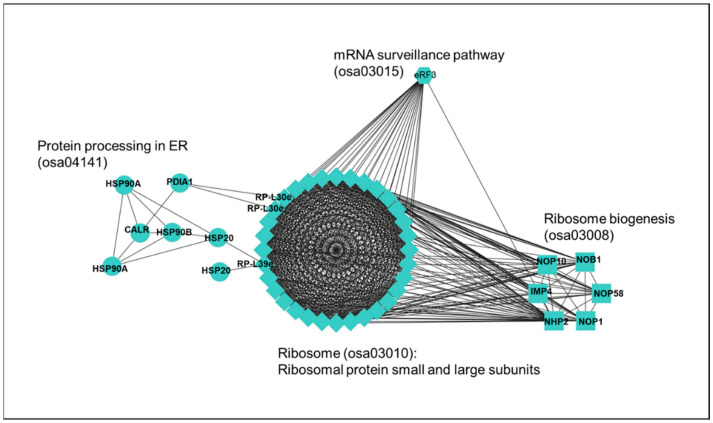
Protein-protein interaction network of candidate genes linked with grain quality. The interaction network among proteins encoded by genes correlated with seed quality was analyzed using the STRING web-based tool (https://string-db.org/cgi accessed on 1 August 2023) and then visualized using Cytoscape software (ver. 3.4.0). Circle, diamond, square, and hexagon shapes indicate respect 1 tively, candidate proteins involved in protein processing ER, Ribosomal protein small and large subunits, ribosome biogenesis, and mRNA surveillance.

**Figure 9 ijms-24-16941-f009:**
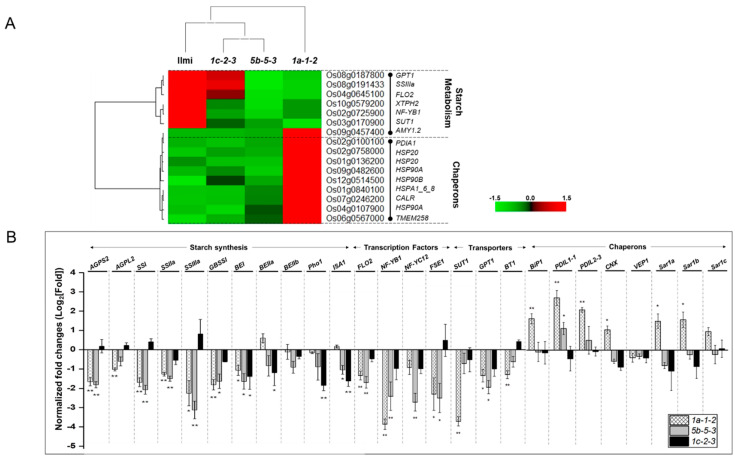
Starch metabolism genes are positively correlated and ER chaperones are negatively correlated to decrease in grain weight in *glutelin* gene-edited lines. (**A**) Heatmap showing altered expression of genes involved in starch metabolism pathway and ER stress pathway in *glutelin* gene-edited lines (*1a-1-2*, *5b-5-3*, and *1c-2-3*) relative to the wild-type (cv. *Ilmi*). The expression level of each gene among different lines was normalized by Z-score. (**B**) Relative expression level of genes involved in starch metabolism pathway and ER stress pathway in *glutelin* gene-edited lines (*1a-1-2*, *5b-5-3*, and *1c-2-3*) using quantitative real-time PCR (q-RT-PCR). Error bars denote ± SD of three replicates. *p* values were calculated by the Student’s *t* test (* *p* < 0.1, ** *p* < 0.01).

**Table 1 ijms-24-16941-t001:** Functional categories of the transcripts deferentially expressed in glutelin gene-edited lines compared with those in Ilmi.

Bin Code ^a^	Annotation	1a-1-2 vs. Ilmi				5b-5-3 vs. Ilmi				1c-2-3 vs. Ilmi				Total Number of Non-Redundant Genes	
		Up		Down		Up		Down		Up		Down			
		No	%	No	%	No	%	No	%	No	%	No	%	No	%
1	Photosyntheisis	18	1.86%	0	0.00%	31	1.39%	0	0.00%	17	2.21%	0	0.00%	34	1.01%
2/3/4/5/6/8 ^b^	Carbohydrate metabolism	7	0.72%	4	2.07%	4	0.18%	8	2.07%	1	0.13%	1	1.14%	20	0.59%
9.	Mitochondrial electron transport/ATP synthesis	5	0.52%	0	0.00%	20	0.90%	2	0.52%	10	1.30%	0	0.00%	23	0.68%
10	Cell wall	6	0.62%	4	2.07%	13	0.58%	5	1.30%	1	0.13%	0	0.00%	27	0.80%
11	Lipid metabolism	1	0.10%	5	2.59%	8	0.36%	10	2.59%	1	0.13%	0	0.00%	20	0.59%
13	Amino acid metabolism	5	0.52%	0	0.00%	7	0.31%	2	0.52%	0	0.00%	0	0.00%	14	0.42%
15/16	Metal handling & secondary metabolism	4	0.41%	1	0.52%	7	0.31%	3	0.78%	1	0.13%	0	0.00%	13	0.39%
17	Hormone metabolism	13	1.34%	2	1.04%	19	0.85%	4	1.04%	2	0.26%	0	0.00%	38	1.13%
18/19	Co-factor/vitamine metabolism and tetrapyrrole synthesis	2	0.21%	0	0.00%	4	0.18%	1	0.26%	0	0.00%	0	0.00%	6	0.18%
20	Stress	24	2.48%	6	3.11%	29	1.30%	13	3.37%	3	0.39%	0	0.00%	65	1.93%
21	Redox	4	0.41%	0	0.00%	6	0.27%	3	0.78%	0	0.00%	1	1.14%	14	0.42%
23/24/25 ^c^	Other metabolism-related BINs	3	0.31%	1	0.52%	2	0.09%	6	1.55%	0	0.00%	0	0.00%	11	0.33%
26	Miscellaneous	16	1.65%	11	5.70%	35	1.57%	17	4.40%	7	0.91%	3	3.41%	70	2.08%
27.1	RNA.processing	110	11.35%	35	18.13%	145	6.51%	36	9.33%	127	16.49%	32	36.36%	258	7.67%
27.3	RNA.regulation of transcription	15	1.55%	6	3.11%	43	1.93%	13	3.37%	7	0.91%	0	0.00%	73	2.17%
27.2/4	RNA.transcription and RNA binding	5	0.52%	0	0.00%	8	0.36%	3	0.78%	1	0.13%	0	0.00%	14	0.42%
28	DNA.synthesis and repair	4	0.41%	2	1.04%	16	0.72%	4	1.04%	5	0.65%	1	1.14%	28	0.83%
29.1/3/6/7/8 ^d^	Protein others	7	0.72%	1	0.52%	9	0.40%	4	1.04%	2	0.26%	2	2.27%	20	0.59%
29.2.1/2/3/4	Protein synthesis-ribosome, initiation, & elongation	49	5.06%	0	0.00%	44	1.98%	4	1.04%	21	2.73%	7	7.95%	83	2.47%
29.2.6	Protein synthesis-ribosomal RNA	12	1.24%	1	0.52%	102	4.58%	2	0.52%	80	10.39%	1	1.14%	113	3.36%
29.2.7	Protein synthesis_transfer RNA	56	5.78%	2	1.04%	129	5.80%	2	0.52%	89	11.56%	3	3.41%	145	4.31%
29.4	Protein postranslational modification	5	0.52%	6	3.11%	21	0.94%	13	3.37%	1	0.13%	0	0.00%	43	1.28%
29.5	Protein.degradation	7	0.72%	9	4.66%	27	1.21%	21	5.44%	2	0.26%	1	1.14%	59	1.75%
30	Signalling	16	1.65%	2	1.04%	22	0.99%	13	3.37%	4	0.52%	0	0.00%	47	1.40%
31	Cell organization	8	0.83%	1	0.52%	15	0.67%	3	0.78%	1	0.13%	2	2.27%	26	0.77%
33	Development	7	0.72%	3	1.55%	15	0.67%	20	5.18%	0	0.00%	1	1.14%	43	1.28%
34	Transport	14	1.44%	7	3.63%	19	0.85%	14	3.63%	1	0.13%	2	2.27%	52	1.55%
35	Unknown	546	56.35%	84	43.52%	1426	64.06%	160	41.45%	386	50.13%	31	35.23%	2009	59.72%
Total		969	100.00%	193	100.00%	2226	100.00%	386	100.00%	770	100.00%	88	100.00%	3364	100.00%

^a^ BIN codes of genes were produced according to MapMan classification using the MapCave tool (http://mapman.gabipd.org/web/guest/mapcave (accessed on 29 July 2023)). ^b^ BIN 2/3/4/5/6/7/8 is carbohydrate metabolism-related BINs: major carbohydrates (BIN 2), minor carbohydrates (BIN 3), glycolysis (BIN 4), fermentation (BIN 5), gluconeogenesis/glyoxylate cycle c c c (BIN 6), OPP cycle (BIN 7), and TCA/organic acid transformation (BIN 8). ^c^ BIN 23/24/25 is other metabolism-related BINs: nucleotide metabolism (BIN 23), biodegradation of xenobiotics (BIN 24), and C1-metabolism (BIN 25). ^d^ BIN 29.1/3/6/7/8 is protein other BINs: protein aa activation (BIN 29.1), protein targeting (BIN 29.3), protein folding (BIN 29.6), protein glycosylation (BIN 29.7), and protein assembly and cofactor ligation (BIN 29.8).

## Data Availability

Data are contained within the article and Appendix A.

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
