# Peer review of "Down-Regulation of Rice Glutelin by CRISPR-Cas9 Gene Editing Decreases Carbohydrate Content and Grain Weight and Modulates Synthesis of Seed Storage Proteins during Seed Maturation"

_ijms, 2023, doi:10.3390/ijms242316941_

Round 1

Reviewer 1 Report

Comments and Suggestions for Authors

This paper investigates the effects of editing mutant rice glutenin subunits using the CRISPR-Cas9 gene on starch content, grain weight, and other seed storage proteins during rice seed maturation. This study observed the reduction of total SSPs and suppression of targeted SSPs in seeds of glutelin gene-edited lines, together with enhanced expression of other glutelin genes and increased number of smaller and irregular PB-IIs, and suggest involvement of post-transcriptional, translation-related, and post-translational mechanisms leading to reduced weight and starch content in seeds from glutelin gene-edited rice.

There are some problems, which must be solved before it is considered for publication.

On page three, the author confirmed the specificity of each sgRNA through in vitro DNA cleavage test. I believe there are some shortcomings in the specificity testing here. The specificity verification is performed using a specific matching sgRNA combination Cas9 nuclease for cleavage, but there is no control experiment to prove that specific sgRNA cannot cleave other target sequences.

On page six, the phrase 'Responding to their altered psychology,' should be combined with the first paragraph on page seven. Here, page seven starts a new paragraph directly, otherwise the phrase ',' on page six will appear abrupt.

Comments on the Quality of English Language

The author should carefully improve the quality of English language through the whole manuscript.

Author Response

All your comments have been thoroughly addressed and the manuscript was revised accordingly. Please refer to the attached file for details.

Reviewer 2 Report

Comments and Suggestions for Authors

Authors (Chandra et al.) have generated the transgenic rice lines with suppressed GluA1, GluB2 and GluC1 genes. The effect on expression of Glu genes is confirmed by qRT-PCR and Western blots. Authors report several lines of evidence in these lines including reduced starch and amylose content, reduced grain weight and irregular shape of protein aggregates in mature seeds as compared to control seeds. They have further performed transcriptional profiling and found positive correlation with qRT-PCR results. Also, KEGG pathway analysis and GO analysis of differentially expressed genes relate to RNA processing and protein translation including ribosome biogenesis, protein processing in ER suggesting the involvement of post-transcriptional and post-transcriptional mechanisms that leads to reduced starch content and reduced weight in glutelin gene-edited transgenic rice seeds.

Comments:

1. Figure 3C: Please provide the data for a loading control e.g., Gapdh.

2. Figure 5A: Please provide the details about number of seeds included for scanning electron microscope experiments for each transgenic line including wild type and whether all of the seeds in each transgenic line had the irregular shape. If not, then please provide the numbers/frequency of the seeds with irregular shape either in the figure legend or the results section.

3. Figure 5B and 5C: Please mention the number of seeds used in these experiments in figure legends.

4. Please mention the criteria for selecting only these 3 lines (1a-1-2, 1c-2-3 and 5b-5-3)? How many transgenic lines had mutations corresponding to 1a-1-2, 1c-2-3 and 5b-5-3?

5. Please provide the total number along with the frequency in figure 2.

6. As mentioned in the method section (line 574-575) that gRNAs authors have used gRNA that targets more than one gene, were there any transgenic line with more than one gene mutated? If yes, how well they performed in terms of growth and what was the effect on seed quality (weight, shape, size, starch content, SSP content, etc.)

7. Line 184-186 and 275-277: Since authors have observed the compensatory up regulation of some other glutelins in the edited seeds in the Western blots, and up regulation of prolamins gene expression in seeds with glutelin knockdown, it would be worth cross-checking and explaining the details of expression of how other glutelin genes are affected in the edited seeds with the knockdown of one specific glutelin gene. 

Author Response

(The authors gave the same response as above.)

Reviewer 3 Report

Comments and Suggestions for Authors

Reviewer comments:

The manuscript ID (ijms-2719791) entitled “Downregulation of rice glutelin by CRISPR-Cas9 gene editing decreases carbohydrate content and grain weight and modulates synthesis of seed storage proteins during seed maturation” by Chandra et al. I found this topic interesting, demonstrates rice glutelin genes that regulate synthesis of starch and seed storage proteins. But I have a few concerns related to the research article. I am asking authors to revise the manuscript carefully considering my comments for possible publication in “International Journal of Molecular Sciences”.

I have given my comments.

• The present investigation will be a good contribution to the genetic improvement of Rice quality.

• Line No 34: The authors requested to remove ‘but’.

• Line No 35 to 36: Authors requested to check and rewrite the sentence ‘Wheat (Triticum aestivum), maize (Zea mays) and rice are important global food staple cereals, at least in part because their seeds contain up to 12% by weight of seed storage proteins (SSPs)’.

• Line No 197: Authors provided the Morphology of seeds but did not mention phenotype of plants.

Authors requested to add genome edited plants image in supplementary figure.

• Line No 677: Authors must check and correct ‘Transcriptope Data Analysis’ with ‘transcriptome’

• Authors should mention the “original deep sequence” of genome edited plants. Here in Fig S2, Fig S3, Fig S4 Glu edited sequences shown in **************.

• The authors are requested to check ‘SpCas9 gene’ size in gel images, each gel image showing different size (Fig S2, Fig S3 and Fig S4).

• Authors are requested to mention what type of ladder/marker used gel images.

The submitted manuscript may be acceptable for publication after a major revision.

Comments on the Quality of English Language

English language editing required especially in introduction part.

Author Response

(The authors gave the same response as above.)

Round 2

Reviewer 1 Report

Comments and Suggestions for Authors

I have no more comments on this manuscript.